# Distributed Blockchain-Based Message Authentication Scheme for Connected Vehicles

**Jaewon Noh** **, Sangil Jeon** and **Sunghyun Cho \***

Department of Computer Science and Engineering, Hanyang University, 55 Hanyangdeahak-ro, Sangnok-gu, Ansan 15588, Gyeonggi-do, Korea; wodnjs1451@hanyang.ac.kr (J.N.); nesi0324@hanyang.ac.kr (S.J.)

**\*** Correspondence: chopro@hanyang.ac.kr; Tel.: +82-31-400-5670

**Abstract:** Vehicular ad-hoc networks (VANETs) have several security issues such as privacy preservation, secure authentication, and system reliability. In the VANET, a vehicle communicates with other vehicles or infrastructures using broadcasting messages. These messages contain not only normal traffic information, but also identification information of sender. In general, the identification information remains encrypted to ensure privacy. However, the conventional centralized system can decrypt the identification information using private information of the sender vehicle. As a result, the central server can often be targeted by adversaries. We propose a message authentication scheme for anonymity and decentralization of information using blockchain technology. Here, we introduce public-private key and message authentication code (MAC) for secure authentication. In this paper, we adopt consensus algorithms for composing blockchain system such as the proof of work (PoW) and Practical Byzantine Fault Tolerance (PBFT) into the proposed authentication process. Finally, we demonstrate that the proposed method is secure from the attacks which include impersonation from internal attacker as well as typical attacks.

**Keywords:** authentication; blockchain; connected vehicle; privacy preservation; consensus

## 1. Introduction

Connected vehicles are driving securely and efficiently, avoiding accidents by communicating with various sensors and infrastructures on wireless environment. Figure 1 shows the architecture of the VANET, which is a subset of mobile ad hoc network (MANET). The conventional VANET consists of trusted authority (TA), base station which connects to the core network, road side units (RSUs), and on-board units (OBUs) equipped in the vehicles. The base station serves as the backbone of the entire system and communicates with the RSUs through secured wired connections. The RSU forwards broadcasted messages or sends information from the center to the vehicles. The OBU performs data processing and broadcasts safety messages to the network. In the VANET, the vehicle uses IEEE 802.11 based dedicated short-range communication (DSRC). There are two representative communication types such as vehicle-to-infrastructures (V2I) and vehicle-to-vehicle (V2V). The vehicular message can contain various information including traffic, navigation, and emergency. There are several security requirements for secure vehicular communication. One important requirement is that the message should not expose the private information of a specific vehicle such as real identity. Though, receivers should be able to prove the validity of the contents like sender information in the message. Many researches related to secure authentication have been conducted in the vehicular networks. However, the existing studies have some limitations. Most of works may cause a bottle-neck and centralization of information in the system. In the centralized architecture, single point of failure problem may occur. If the central entity is attacked, stored data would be used maliciously. In addition, several attacks such as impersonation and forgery from insiders in the network. These problems

are not resolved completely and should be solved. Therefore, we propose improved authentication scheme for connected vehicles by adopting blockchain technology to compose decentralized vehicular network. There are three primary benefits by adopting blockchain technology in the connected vehicle network. First, the sharing of transactions or blocks mitigates the burden of centralized entity. Second, the distributed connected vehicles are capable of performing anonymous authentication, even within an unreliable environment. Finally, every vehicles can access the block and can check the guaranteed integrity of the message without help from a central system. Our contributions are as follows.

- We propose distributed message authentication scheme based on blockchain, the vehicles can authenticate the broadcast messages in distributed manner.
- In our system, we consider malicious nodes which exist in the network. The proposed scheme can prevent several attacks from the insiders.
- We provide formal verification for the proposed scheme and implement the proposed system which the mobility of vehicles is considered.

The remainder of the paper is organized as follows. In Section 2, the related works are described. Section 3 introduces background of blockchain technology. Section 4 describes our system model including network model, block structure, adversary model, and system goals. The proposed authentication scheme is described in Section 5. In Section 6, we analyze security and overhead of the proposed authentication scheme. Finally, the conclusions are described in Section 7.

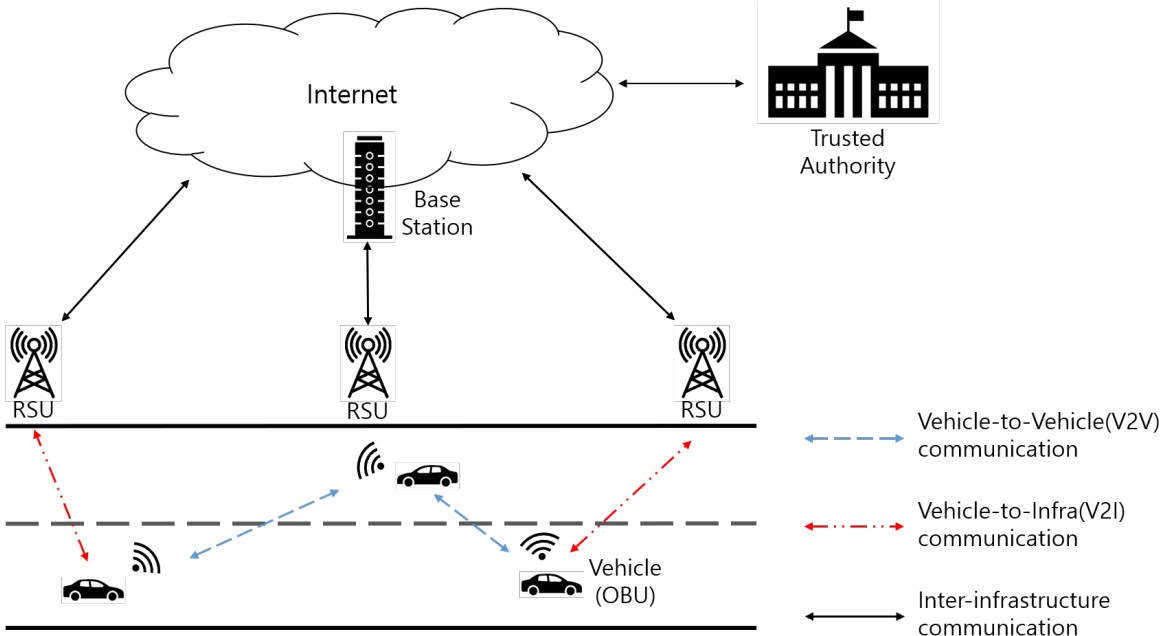

**Figure 1.** Vehicle ad hoc network architecture.

## 2. Related Works

Recently, lots of researches have been performed for improving security in the vehicular network. Commonly, confidentiality, integrity and authenticity are important issues for secure vehicular network. In addition, the researches which assure privacy are another major issue. The authentication and location-based service are representative targets which require privacy preservation. In the [1,2], they described security challenges, requirements and attack methods. The conventional approaches utilized the elliptic curve digital signature algorithm (ECDSA) and timed efficient stream loss-tolerant authentication (TESLA). While the ECDSA can reduce resources used by providing strong security with shorter bits, it does not ensure non-repudiation and it requires high computational cost. The TESLA is a symmetric protocol which reduces computational complexity. However, it stores all data until the

symmetric key arrives. This makes the system vulnerable to the memory attack. The TESLA++ is the enhanced version of the TESLA. It reduced memory consumption by storing self-generated message authentication codes. Nonetheless, both ECDSA and TESLA++ cannot guarantee the non-repudiation and privacy preservation of message senders. Recent researches on secure VANET communication can be divided into three types: pseudo authentication, group, and hybrid schemes [3]. The pseudo authentication scheme uses public key infrastructure (PKI) and pseudonymous certificates for signing messages and verification. In the pseudo authentication scheme, several issues for managing certificates or generating pseudonyms should be included. The group scheme relies on groups to hide members from the adversary, where group members have a secure channel with a group key. In this case, an adversary finds it difficult to penetrate communication line. However, the grouping issues should be handled such as selecting header, joining and leaving. The hybrid scheme utilizes both pseudo authentication and group schemes. In the [4], the authors proposed hybrid scheme which uses both ECDSA and the TESLA++. Unfortunately, this scheme cannot guarantee strong privacy preservation, because their system cannot keep the vehicle's identity information secure against hacking attempts. Other researches [3,5,6] focused on addressing identity privacy issues by introducing PKI and tamper-proof devices to prevent the release of unencrypted identity information. They tried to solve various privacy problems commonly found in VANETs with assuring message integrity and non-repudiation. In our previous work [6], we introduced both the public key infrastructure and an authentication table. The authentication table, which contains identification information, is kept by the local trusted authority (LTA). The LTAs are deployed in their regions and authenticate vehicles instead of a central trusted authority. In short, all messages broadcasted in each region should pass through corresponding LTA for verification process. In addition, Ref [7] categorised authentication schemes in VANET and analyzed properties of the conventional schemes. The schemes are classified depending on the approaches such as cryptography, sign and verification. Each scheme has advantages and limitations. For privacy preservation, various attacks such as identity disclosure, impersonation, repudiation, tracing and insider should be considered. However, most conventional schemes are centralized and cannot solve above problems completely. For example, in the group signature-based authentication scheme, they should share a specific key for clustering, and generate the key using the keys of the all group members [8]. Batch verification scheme can verify multiple messages at a time but it may fail to verify it when a few messages are forged [9]. The center or a cluster head takes the process as a representative. The other authentication schemes which are based on cryptographical approaches are important to improve efficiency such as computation complexity, communication overheads with assuring security requirements [10,11]. Thus, decentralization is one of solutions to enhance communication efficiency in the conventional vehicular network. On the other hand, several researches have considered blockchain in the vehicular network. The authors in [12] adopted blockchain to VANET for message dissemination, especially critical event information. They tried to ensure integrity of the event messages by storing in the blockchain. However, their scheme did not consider various attacks caused from the wireless environment. They used blockchain just as a distributed ledger. In [13], the authors introduced blockchain technology which is one of the distributed solutions. They show the possibility that the blockchain can be used for improving security and privacy for automotive environment. Tracking malicious node and assuring anonymity are possible applications using blockchain. Also, blockchain-based network can ensure integrity of messages through consensus between members. Thus, blockchain is a suitable option for security-critical network, especially connected vehicular network. From the blockchain-based architecture, the network can be decentralized with assuring message integrity through consensus among the network members. The recent blockchain-based authentication scheme tried to solve centralized architecture, anonymity and trust characteristics [14]. They proposed lightweight authentication mechanism in vehicular fog infrastructure. In their scheme, entities perform cross-datacenter authentication and key exchange based on blockchain. In this paper, we solve the centralization and privacy exposure problems in vehicular communication. When the vehicles broadcast safety messages, the messages should be

authenticated whether it is not forged. In the existing schemes, the TA authenticates the authorized vehicles or the vehicles have to remain information about the neighbor vehicles. This may cause privacy exposure. To solve the problems, we use blockchain for ensuring anonymity, unforgability, and traceability with decentralized architecture. In the proposed system, the central entity is not involved for authentication or messaging, except initial procedure. The other entities can broadcast and authenticate messages in distributed manner.

## 3. Preliminary Knowledge

### 3.1. Blockchain

The blockchain is one of the emerging technologies, and has been actively researched in various fields, including industrial technology, security, and finance [15–17]. Most of centralized systems like data management and banking system suffer from malicious attacks. Thus, the systems should guarantee complete security against the attacks. If a central node is compromised, it would tamper lots of information. Blockchain technology can prevent the problem by sharing same blocks. Blockchain consists of a set of blocks connected by hash values. All blocks possess the hash value of the immediately preceding block except the first block called genesis block. In the blockchain-based system, multiple nodes participate to generate and verify a block through consensus. A generated block is broadcasted in the network. Therefore, the system can be decentralized without central administrator, allowing participants in the network to access the shared ledger where the blocks are stored. Through those blocks, participants can observe records of transactions whenever they want. Blockchain networks are classified by public, private and consortium depending on the structures. While anyone can be a node in the public blockchain network, permission from the system administrator is required in the private blockchain network. Consortium blockchain is mixed architecture of both public and private blockchain.

### 3.2. Consensus Algorithm

The consensus algorithm is the most important part in the blockchain technology. It enables distributed nodes to maintain the same blockchain. The main purpose of the consensus algorithm is to assure each node to verify block generation in a distributed manner. There are three typical consensus algorithms, proof of work (PoW), proof of stake (PoS), and Practical Byzantine Fault Tolerance (PBFT) [18,19]. The PoW consensus algorithm is the most widely used consensus algorithm in public blockchains. In the PoW based network, a miner who wants to generate a block should find a specific hash value. In order to find the value, every miners compete based on their hash power. In contrast with the PoW consensus algorithm, the miner who stakes coin in wallet software can generate blocks in the PoS-based system. Both PoW and PoS consensus algorithms have a problem. A miner who has large hash power available for calculation, or possesses large amounts of stakes, can affect the block generation. However, the PBFT consensus algorithm uses a two-thirds majority voting method. It is mainly used in private blockchain, and IBM hyperledger fabric is a typical example. The procedure of the PBFT consensus algorithm consists of REQUEST, PRE-PREPARE, PREPARE, COMMIT, and REPLY. Figure 2 shows the overall procedure of the PBFT algorithm. In the PBFT-based system, every node can access the public key of other participants. Therefore, each node can determine the sender of a transaction. There are two node types, a primary and replica nodes. The primary node is selected from the leader selection process or when it receives a message from the client first. The replica node is the other nodes in a cluster, except the primary node. During the request process, the client broadcasts a transaction to all nodes. The primary node generates a block with the received transactions and broadcasts the block in the pre-prepare process. In the prepare and commit process, each replica node confirms whether they have received the same block or not. Then, they verify that the transactions and values in the block are correct. Finally, all nodes send the verification result to the client in a reply process.

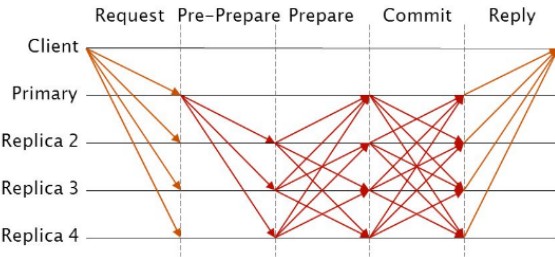

**Figure 2.** Procedure of PBFT consensus algorithm.

## 4. System Model

In this section, we describe our system model, including the network model, block model, adversary model, and system goal. The system model is based on the [3,5,6]. We consider four system goals: strong privacy preservation, message integrity, non-repudiation and traceability.

### 4.1. Network Model

Our network model consists of a root trusted authority (RTA), LTAs, RSUs and vehicles as shown in the Figure 3. They compose private blockchain network and share blocks. The RTA manages the entire system and is trusted by all entities because only authorized vehicles belong to the network. However, the RTA leans out authenticating vehicles. It just takes charge of registration and traces a malicious user when abnormal actions are reported. Each LTA is responsible for authentication and is mutually authenticated using its own signature by the vehicles in its area. In each region of LTAs, a system key is shared. Both RTA and LTAs have sufficient resources and computing power, and they can communicate with each other directly. Biological identification information is stored securely in the RTA after the registration. The RSU forwards messages between vehicles and the LTA. The OBU, in the vehicle, performs data processing. The vehicles have sufficient storage capacity to store blockchain. Finally, the driver can use a biometric information (BI) such as fingerprints for authentication.

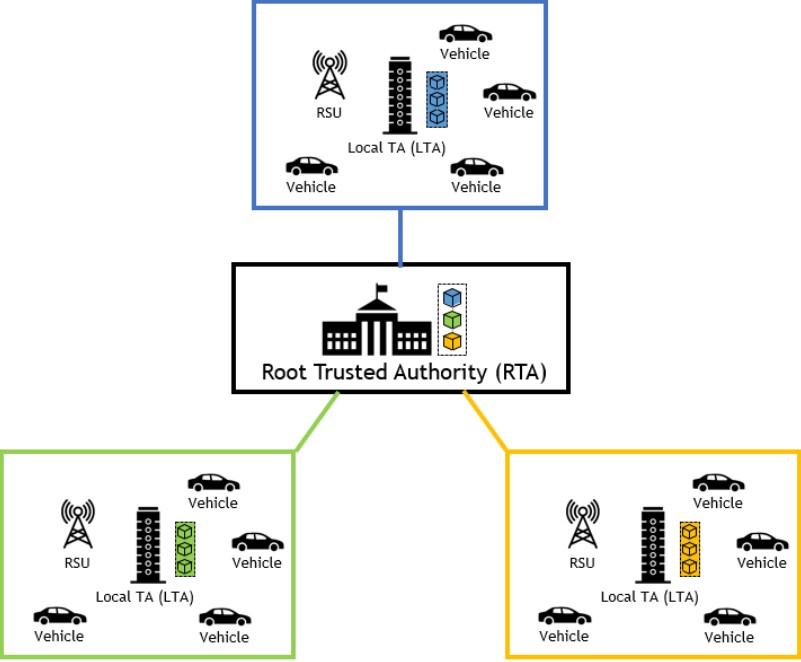

**Figure 3.** Considering network model.

*4.2. Block Model*

In this paper, we consider the messages in the vehicular network as blockchain transactions. Vehicles and infrastructures communicate with each other by sending messages. The abstract blockchain model is shown in the Figure 4. Each block is connected by its hash value. The block header contains the hash value of the previous block. Therefore, we can trace the order of block generation through the block header. In addition, we can validate a message through the merklehash which is summarized hash value of all messages in the block. The process of merklehash generation is described in Figure 5. Merklehash is computed using hash values of individual messages in the block. Therefore, we can determine whether the transaction is in the block. The block header consists of the software version, generation time, previous block hash, merklehash, nonce, and difficulty. The previous block hash refers to the hash value of the preceding block and the merklehash refers to the result of computed hash values from the all messages in a block. Difficulty refers to the hash difficulty which affects the block generation time. The blocks are generated continuously and are broadcasted in the network. The vehicle can obtain block information from other vehicles or an LTA when the vehicle needs to check blockchain. In our scheme, both PoW and PBFT consensus algorithms are used. In order to make distributed architecture, the LTA only generates new block and doesn't participate in block verification. After the block verification, an RTA and LTAs confirm whether the correct block is received or not.

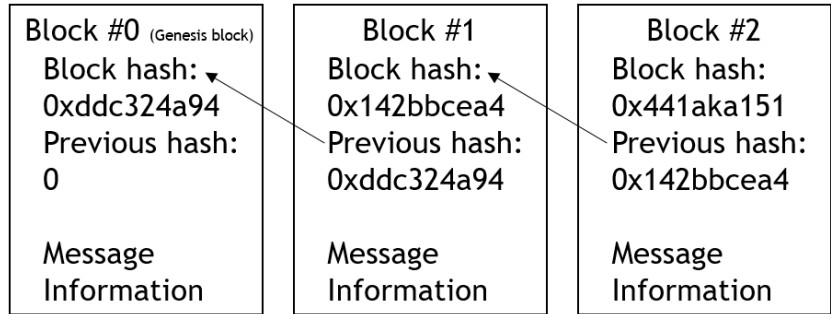

**Figure 4.** Abstract blockchain structure.

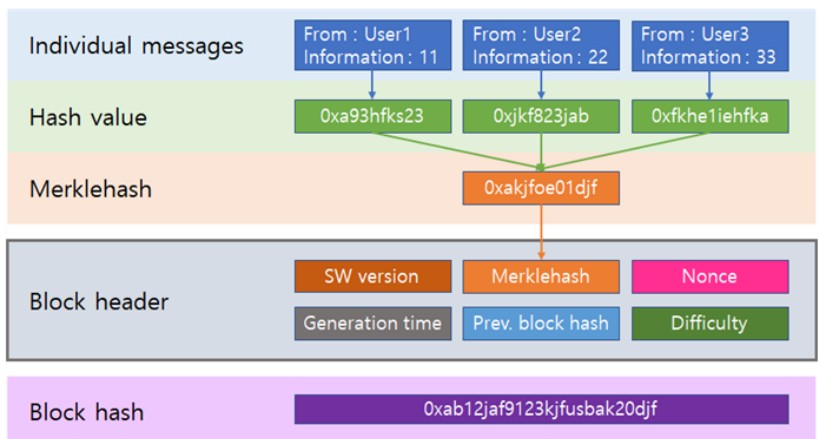

**Figure 5.** Detailed block structure.

*4.3. Adversary Model*

Adversaries can participate in communication channels with the other vehicles, as well as send and receive messages with the vehicles. In addition, it has powerful communication equipment with sufficient resources. This allows the adversary to attempt two types of attacks: passive and active. The passive attacks include sniffing, which does not interrupt or tamper the target, while active attacks

affect the target directly by forging information (for example, man-in-the-middle attack). Especially, we must consider internal adversary because any node can be an adversary. Compare with the existing adversaries, internal adversaries are authorized because they are registered in the network. Internal attacks can occur from the authorized insiders, and they have a shared system key. Therefore, the adversaries can perform various attacks such as impersonating other vehicles by using the shared key or sending malicious messages.

*4.4. System Goal*

### 4.4.1. Strong Privacy Preservation

Strong privacy preservation means that user identification information should be protected against a variety of malicious attacks. The message should not contain unencrypted identity of the entities. Therefore, the adversary cannot obtain the identification information of the vehicle, even if they succeed message eavesdropping. For example, if a system like the LTA is hacked, the identification information should still be kept securely.

### 4.4.2. Message Integrity

The adversary may attempt to send a message with modified information. If there is no way to discern the correct message, integrity cannot be assured. Therefore, a verified message should be guaranteed tamper-proof.

### 4.4.3. Non-Repudiation

A sender must not be able to deny the fact that it sent a message. Through the verification process, only messages from the correct sender should be delivered. Therefore, only verified messages remain in the block.

### 4.4.4. Traceability

The vehicles have to communicate with other vehicles with anonymity to avoid tracking by the adversary. However, the administrator like RTA should be able to trace the adversary when malicious actions occur.

## 5. The Proposed Authentication Scheme

In this section, we describe the proposed authentication process. The entire process is shown in Figure 6. The proposed blockchain-based authentication scheme consists of six phases: initiation, registration, message sign, message verification, block generation, and block confirmation. We divide nodes into block generation node and verification node. The infrastructure nodes have higher computing power, so they are in charge of block generation, especially through PoW consensus. The vehicles have relatively lower computing power and consensus should be completed quickly because of the mobility. Thus, the PBFT consensus is used in the verification. Consequently, block generation is performed by proof-based algorithm, and block generation is done by voting-based algorithm. The Table 1 describes the notations and description which are used in the proposed scheme.

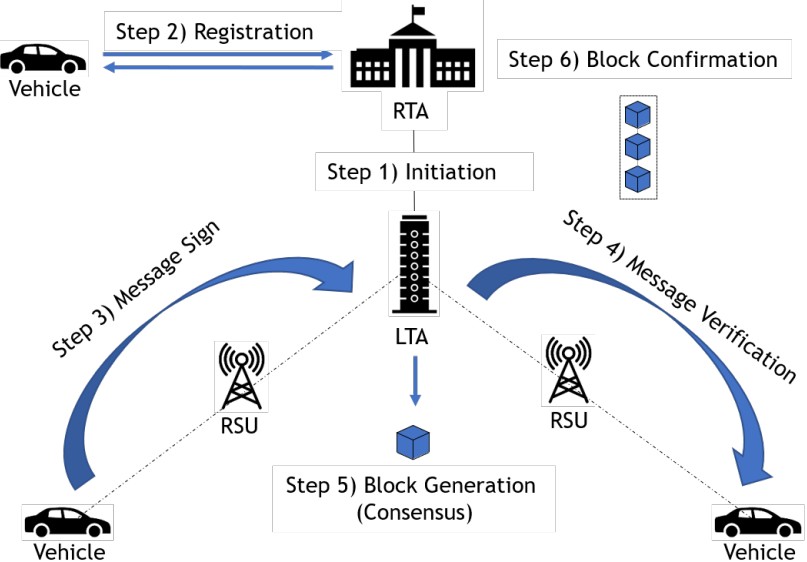

**Figure 6.** Entire message authentication process.

### 5.1. Initiation

The RTA generates its own public and private asymmetric keys. A message encrypted with the public key can only be decrypted with the private key. The opposite is also possible. The RTA generates the genesis block, system key *k* and the list of LTAs. The block containing the previous block hash value for zero is the genesis block.

### 5.2. Registration

Vehicles who want to join the network should visit the RTA first. The driver gives his vehicle information and BI to the RTA. The RTA generates and gives a public-private key set for the vehicle and the system key *k*. The vehicle can use own public-private key in the set. Therefore, the identification information is kept securely. The LTA receives genesis block, system key *k* and the public key of RTA. The vehicle registration process is shown in Figure 7a and the LTA registration process is shown in Figure 7b respectively. After the registration process, registered vehicle is authorized in the vehicular blockchain network.

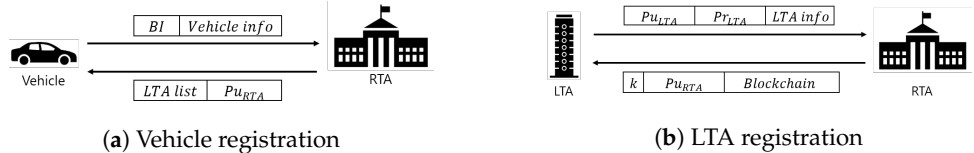

(**a**) Vehicle registration　　　　　　　　(**b**) LTA registration

**Figure 7.** Registration processes.

### 5.3. Message Sign

A vehicle authenticates BI information of the driver. If it is matched with the registered authentication information, the vehicle can broadcast messages. The LTA generates message authentication code (MAC) $mac_k(m)$ and broadcasts $Enc_k(Enc_{Pr_{LTA}}(prevhash||h(m')))$ with the MAC. The registered vehicles which has *k* derive *prevhash* from the sent message using the public key of the LTA. Therefore, the vehicle which does not have the public key of LTA and the system key cannot participate in communication. The vehicle generates message *m* and computes its hash value $h(m)$. The vehicle concatenates previous hash values of blocks and $h(m)$, then encrypts those with its private key, as described in Equation (1). The vehicle signs with the system key *k* and broadcasts it to the network, as described in Equation (2). The Figure 8 shows the format of the message.

$$Enc_{Pr_{Vehicle_i}}(prevhash||h(m)) \tag{1}$$

$$Enc_k(Enc_{Pr_{Vehicle_i}}(prevhash||h(m)), m, ts), mac_k(m) \tag{2}$$

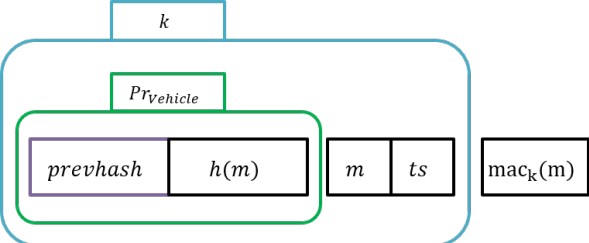

**Figure 8.** Message format.

### 5.4. Message Verification

When vehicles and LTA receive messages, they perform verification process as follows. The vehicles check the MAC from the received message with system key $k$. Then, the vehicle decrypts the message with the public key of the sender. All vehicles in the same LTA area can know each other's public key from broadcasting. Vehicles then determine if the received hash value of the previous block is equivalent with the prevhash value from previous block. The vehicle then compares the received $h(m)$ and the calculated hash value, which are computed from the received $m$. The process of message decryption is shown below.

### 5.5. Block Generation and Verification

While the V2V messages are exchanged, an LTA collect the messages. In certain period, the LTA generates a block based on the PoW consensus. After the block is generated, the block is verified through the PBFT consensus. Then, the block can be connected to the blockchain if it passes the block generation and block confirmation process. The block generation process consists of a generation process and verification process. Once the block has been created, the verification process is mandatory since the contents of the block should be verified. The block generation and verification processes are performed among the vehicles and the LTA, but the block confirmation process is performed among LTAs and the RTA. We introduce the PoW consensus algorithm to our block generation process and the PBFT consensus algorithm to our block verification process. Figure 9 describes the two processes in detail. The block generation process consists of message broadcasting and block information broadcasting. In the block generation process, vehicles continuously broadcast messages until the message broadcasting process ends and the LTA stores all messages. Then, the LTA broadcasts necessary information to make a block, including software version, merklehash, generation time, hash value of previous block, and difficulty. After the information broadcasted to the network, the LTAs try to find hash value which satisfies the specific nonce. The process of finding the nonce is described in Figure 10. The block verification process consists of commit and reply. If a block is generated from the block generation process, the LTA initiates block verification by broadcasting the block. The vehicles become replica nodes and firstly check the signature in the block using their LTA list. If the LTA is authenticated, they believe the same block is broadcasted in the network. Thus, the vehicles can verify commit process only in this case. However, if the vehicles cannot authenticate LTA, then the vehicles should check whether they have same block or not. Like a prepare process, each vehicle determines if they have received the same block as other vehicles by sharing received block. Vehicles can guess the number of the participants by counting prepare messages. During the commit process, the vehicle verifies that the block contains the right merklehash and nonce. If the vehicle verifies enough prepare messages, the vehicle would send commit message about the block. If the

primary vehicle receives more than two thirds of the total number of commit messages, it sends reply message to the LTA. Then block confirmation process is performed between LTAs.

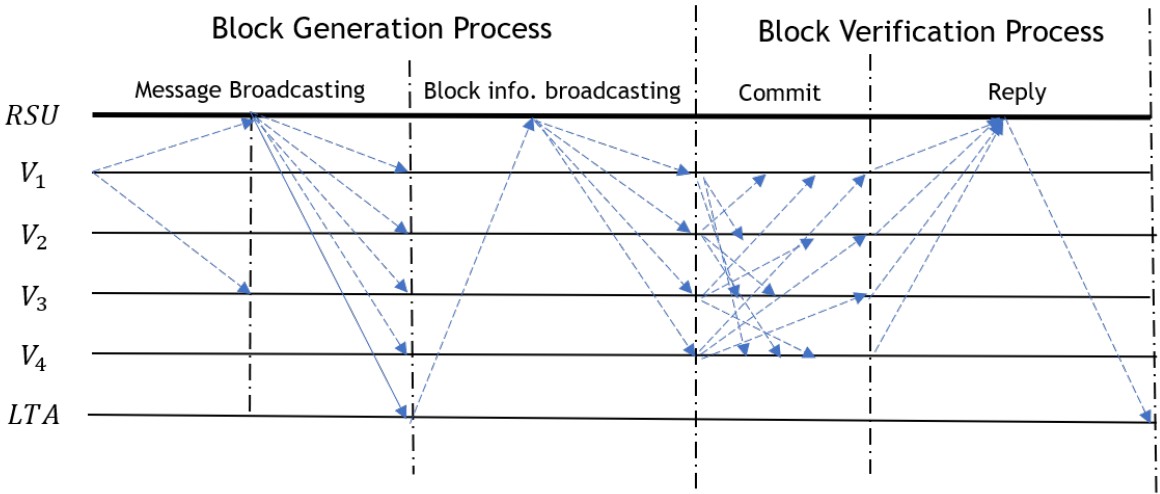

**Figure 9.** Block generation and verification process.

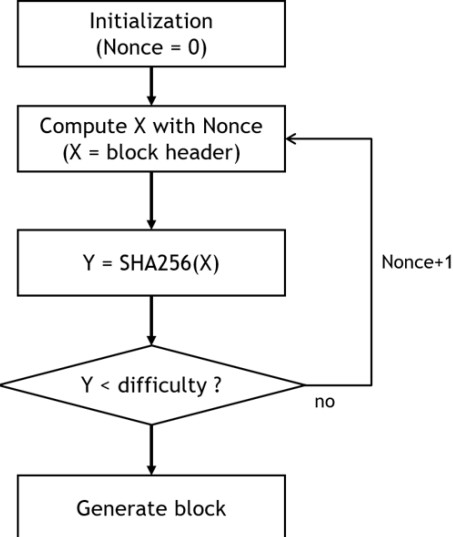

**Figure 10.** Nonce computation process.

*5.6. Block Confirmation*

Once a block is verified, an LTA shares the block to the other LTAs and an RTA. After the confirmation process, the RTA receives the confirmed block from the other LTAs. The block confirmation process is described in Figure 11. The block confirmation process consists of block broadcasting, block confirmation, and reply. This process is performed via wired communication, therefore it is finished much quicker than the previous verification process. The LTA can broadcast a block which has received votes totaling more than two thirds of the total number of vehicles in its area. The LTA sends the block to the other LTAs, and they verify the block with the information contained in the block header. After the verification, each LTA sends result of the verification. If one LTA receives two third confirmation messages of the total number of the LTAs, the LTA sends reply message to the RTA. Then the RTA checks reply messages and connects it to the blockchain finally. From the shared blockchain, all connected vehicles can check records of previous messages. Consequently, the vehicles authenticate messages in distributed manner without any help of the center like RTA. The vehicles can determine whether they believe newly broadcasted messages or not, depending on the blockchain.

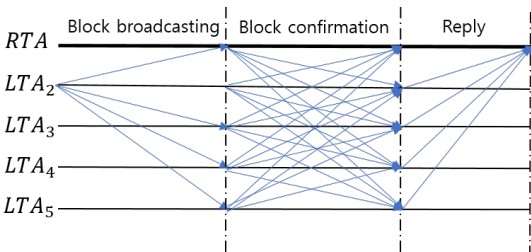

**Figure 11.** Block confirmation process.

## 6. Security and Overhead Analysis

In this section, we analyze the proposed authentication scheme using formal verification method. Besides, we also analyze the proposed scheme about security requirements: strong privacy preservation, message integrity, non-repudiation and traceability. On the other hand, in the block generation process, blockchain cannot be tampered until the number of attackers is less than one half of the total, based on assumptions of the PoW consensus algorithm. Likewise, the block verification process and block confirmation process, which adopt the PBFT consensus algorithm, are secured until the number of attackers is less than one third of the total. Although the compromised entities exist in the network and try to interrupt consensus, they cannot affect result of the consensus algorithm easily, for example failure of the block generation. Thus, the proposed authentication scheme can prevent single point of failure problem. Most of the existing blockchain-based solutions did not handle several issues in detail such as compromised entities or consensus algorithm with strong privacy preservation. We also analyze overheads caused by the proposed authentication scheme by implementation. In the overhead analysis, we consider mobility of the vehicles. We implement vehicular communication system with mobility model. The packet error may occur due to quality of the V2V communication channel. The quality of the channel is affected by distances between vehicles. Thus, we analyze the proposed system based on various measures.

### 6.1. Formal Verification

In this section, we verify correctness of the proposed scheme formally. We use Burrows, Abadi, and Needham (BAN) logic which is one of verification methods [20]. Several notations which are used for the BAN logic are described in the Table 2. In addition, we add some notations ($|\overset{k}{\rightarrow} P$ and $|\not\equiv$) for more expressions in the proposed scheme. Verification using the BAN logic is performed with four steps. First of all, we define the idealized form of the messages in the proposed scheme. In this step, we can remove the unnecessary terms or messages which do not affect security goals. Especially, we select two messages for verification below. Entity can be an LTA or a vehicle.

I1. $LTA_j \rightarrow vehicle_i$: $m_1 = \{Enc_{k_j}(Enc_{Pr_{LTA_j}}(prevhash||h(m))||m||ts), mac_{k_j}(m)\}$
I2. $vehicle_i \rightarrow entity_j$: $m_2 = \{Enc_{k_j}(Enc_{Pr_{Vehicle_i}}(prevhash||h(m))||m||ts), mac_{k_j}(m)\}$

### 6.1.1. Assumptions and Goals

After the idealization, assumptions and goals should be defined. Assumptions for LTAs and vehicles are described below. Most of the assumptions are about relationship between entities and keys. The assumptions are fundamental assumptions which make the proposed system work well. In the system, each entity has own keys and cannot tamper timestamp. The LTAs can generate their own public-private keys after the registration. Also they have a system key and public-private keys of vehicles in their region. Vehicles have also similar assumptions with the LTAs.

A1. $LTA_i |\equiv \#(ts)$
A2. $LTA_i |\Rightarrow (Pr_{LTA_i}, Pu_{LTA_i})$
A3. $LTA_i |\Rightarrow k_{LTA_i}$

A4. $LTA_i \triangleleft Pu_{vehicle_j}$
A4. $LTA_i \mathrel{|\equiv} vehicle_j \mathrel{|\equiv} k_{LTA_i}$
A5. $vehicle_i \mathrel{|\equiv} \#(ts)$
A6. $vehicle_i \mathrel{|\Rightarrow} (Pr_{vehicle_i}, Pu_{vehicle_i})$
A7. $vehicle_i \triangleleft k_{LTA_j}$
A8. $vehicle_i \mathrel{|\equiv} LTA_j \mathrel{|\sim} Enc_{Pr_{LTA_j}}(M)$

On the other hand, there are several goals in below. The goals are related with security requirements, so the goals should be assured in the proposed system. First of all, the messages between the LTA and the vehicle such as $m_1$ and $m_2$, must be trusted. If the malicious vehicle impersonate the other vehicle, a receiver could find contradiction.

G1. $vehicle_i \mathrel{|\equiv} vehicle_j \mathrel{|\sim} Enc_{Pr_{vehicle_j}}(X)$
G2. $vehicle_i \mathrel{|\equiv} LTA_j \mathrel{|\sim} m_1$
G3. $LTA_j \mathrel{|\equiv} vehicle_i \mathrel{|\sim} m_2$
G4. $vehicle_i \mathrel{|\not\equiv} vehicle_j \mathrel{|\sim} \{Enc_{Pr_{vehicle_k}}(X), mac_k(X')\}$

### 6.1.2. Verification

In this section, we verify that the proposed scheme can satisfy the goals as defined above. First, we set the hypotheses and prove them with the rules of the BAN logic.

**Theorem 1.** *Vehicle i believes vehicle j sent $Enc_{Pr_{v_j}}(X)$.*

**Proof of Theorem 1.** Vehicle $j$ uses $Pr_{vehicle_j}$ as a secret key. So, it can only control and generate $Enc_{Pr_{vehicle_j}}(X)$. Vehicle $i$ has $Pu_{vehicle_j}$ and receives $Enc_{Pr_{vehicle_j}}(X)$. Thus, vehicle believes that the received $Enc_{Pr_{vehicle_j}}(X)$ is from the vehicle $j$.

$$V1: \frac{\dfrac{\mathrel{|}\xrightarrow{Pr_{vehicle_j}} vehicle_j}{vehicle_j \mathrel{|\Rightarrow} Enc_{Pr_{vehicle_j}}(X)}}{\dfrac{vehicle_i \triangleleft Pu_{vehicle_j}, v_i \triangleleft Enc_{Pr_{vehicle_j}}(X)}{v_i \mathrel{|\equiv} v_j \mathrel{|\sim} Enc_{Pr_{vehicle_j}}(X)}}$$

□

**Theorem 2.** *Vehicle i believes LTA j sent $m_1$*

**Proof of Theorem 2.** Vehicle $i$ has system key $k_j$ which means $k_{LTA_j}$. Thus, vehicle $i$ has $Enc_{Pr_{LTA_j}}(prevhash||h(m)), m, ts$ from the $m_1$. It can get $prevhash$ and $h(m)$, then believe $m$ if the hash value is same. It is because the vehicle $i$ has also $Pu_{LTA_j}$ and believes the freshness of the time stamp. Consequently, the vehicle $i$ believes LTA $j$ sent the message $m_1$.

$$V2: \frac{\dfrac{\dfrac{vehicle_i \triangleleft k_j}{vehicle_i \triangleleft Enc_{Pr_{LTA_j}}(prevhash||h(m)), m, ts}}{vehicle_i \triangleleft Pu_{LTA_j}, vehicle_i \mathrel{|\equiv} \#(ts)}}{\dfrac{vehicle_i \triangleleft prevhash, h(m), vehicle_i \mathrel{|\equiv} m}{vehicle_i \mathrel{|\equiv} LTA_j \mathrel{|\sim} m_1}}$$

□

**Theorem 3.** *LTA j believes vehicle i sent $m_2$.*

**Proof of Theorem 3.** This proof is similar with the Theorem 2. $LTA_j$ completely controls $k_j$. When the $LTA_j$ receives $m_2$, it can derive $Enc_{Pr_{vehicle_i}}(prevhash||h(m)), m, ts)$. The $LTA_j$ has $Pu_{vehicle_i}$ and believes the freshness of the time stamp. After the $LTA_j$ gets $prevhash$ and $h(m)$, it believes m is not forged. Consequently, $LTA_j$ believes vehicle $j$ sent $m_2$.

$$V3: \frac{\dfrac{\dfrac{\dfrac{LTA_j \mid\Rightarrow k_j}{LTA_j \lhd Enc_{Pr_{vehicle_i}}(prevhash||h(m)), m, ts)}}{LTA_j \lhd Pu_{vehicle_i}, LTA_j \mid\equiv \#(ts)}}{LTA_j \lhd prevhash, h(m), LTA_j \mid\equiv m}}{LTA_j \mid\equiv vehicle_i \mid\sim m_2}$$

□

**Theorem 4.** *Vehicle i don't believe vehicle j sent $Enc_{Pr_{vehicle_k}}(X'), mac_n(X')$ where the vehicles are in the region of $LTA_n$.*

**Proof of Theorem 4.** Let assume Theorem 4 is false, then it can expressed to $vehicle_i$ believes $vehicle_j$ sent $Enc_{Pr_{vehicle_k}}(X), mac_n(X')$. From the V4, vehicle $i$ realizes the sent message is forged. In other word, vehicle $j$ try to forge message by impersonating vehicle $k$. However, these attacks are not possible if malicious attackers have the private key of the target. Consequently, vehicle $i$ don't believe vehicle $j$ sent $Enc_{Pr_{vehicle_k}}(X'), mac_n(X')$ because the assumption is false.

$$V4: \frac{\dfrac{\dfrac{vehicle_i \lhd Pr_{vehicle_k}, k_n, vehicle_i \lhd Enc_{Pr_{vehicle_k}}}{v_i \lhd X}}{vehicle_i \mid\equiv vehicle_k \mid\Rightarrow Pr_{vehicle_k}}}{vehicle_i \mid\equiv vehicle_k \mid\sim Enc_{Pr_{vehicle_k}}(X)}, mac_{k_n}(X')$$

□

*6.2. Security Requirements*

6.2.1. Strong Privacy Preservation

We assumed two situations when the RSU or the LTA are compromised. Although several entities are hacked, the system can assure privacy, especially the real identities of connected vehicles. The real identities are stored only when the vehicles register through the RTA. Multiple hacked RSUs cannot get the information since the RSUs are only responsible for message forwarding. Even if multiple LTAs are hacked, the real biological identities of the drivers are kept safe. The LTA stores the blocks which contain information encrypted with the public key of each vehicle. Therefore, the adversary cannot derive the true biological identity of the drivers without help from the RTA. In our system, multiple adversaries cannot get private key or the real identity of a target vehicle without private key of the target or cooperation from the RTA.

6.2.2. Message Integrity

In the formal verification, $m_1$ and $m_2$ are believed. Through the V2 and V3, it is true that a vehicle knows an LTA sent $m_1$ and the LTA knows the vehicle sent $m_2$, If a message is modified as described in the proof of Theorem 4, the check process can filter the modified message like V4. If the adversary desires for their modified messages to be accepted, they must have the private key of the vehicle because both vehicles and LTA only accept messages encrypted with system key $k$ as well as the

private key of the vehicle. On the other hand, the internal adversary can participate in the network and send messages using the system key *k*. However, all messages are encrypted using the private key of the vehicle, therefore it is difficult for the adversary to send modified messages. Consequently, the integrity of the messages in the network is assured in the proposed system.

### 6.2.3. Non-Repudiation

Repudiation attack is possible when a receiver cannot judge sender of a message. In our system, the BI is used for generating public-private key of the owner and stored only in the RTA. The messages that are normally accepted in the system are encrypted with a private key of the vehicle. Thus, the adversary should derive private key of the target vehicle to forge messages. The internal adversary can attempt to send a malicious message by impersonating the other vehicles by collecting messages in the network, but it cannot be succeeded without the signature using the private key of the target vehicle. Not only the record is stored in the blockchain system, but also the adversary cannot forge the signature of others as shown in the V4. By preventing impersonation, the proposed scheme can assure non-repudiation. If the adversary tries to these kinds of attack, the RTA can identify which vehicle sent the malicious messages using the contents of the block.

### 6.2.4. Traceability

When a malicious action occurs, the system should be able to find the attacker. In the blockchain-based system, normally accepted messages are stored in the block. Therefore, the system can determine the source of malicious messages during the verification phase or confirmation phase. Although the malicious messages are accepted and stored in the blockchain, the contents are verified among the nodes. The only RTA can determine the identity of the adversary utilizing its stored registration information, which includes biological identity and driver information. Therefore, the RTA can assure traceability when the malicious actions are detected in the system.

### *6.3. Overhead Analysis*

Compare with the conventional authentication schemes, the proposed scheme has an overhead caused from the blockchain architecture. Especially, the processing time for consensus is a major overhead. In order to analyze the overhead, we implement the vehicular network under urban street model. We compare the overhead of the proposed scheme with two conventional algorithms such as loof-fault tolerance (LFT) and Hotstuff for comparison. The LFT consensus is the improved version of the PBFT to reduce communication overhead [21]. It simplifies leader selection process and reduces one step compared with the PBFT. At first a leader generates a block and deliver it to the others. Then, the nodes disseminate the block and their vote result. After counting votes, the node commits previous block and broadcasts a new block at the same time. The Hotstuff is a leader-based BFT replication protocol [22]. The Facebook's LibraBFT is one of the examples which uses Hotstuff consensus. In the Hotstuff consensus, there are four phases such as prepare, pre-commit, commit and decide. A header node gathers the messages from the replica nodes and broadcasts next phase messages to the replica node like a communication in star topology. To evaluate consensus delay in the vehicular network, we firstly design vehicular network. Figure 12 shows the simulation model implemented with Python 3. A triangle, located in the center, represents an RSU and red circles represent vehicles. The vehicles move randomly at a speed of 20 m/s in the Manhattan grid model. The vehicles, which are included in the circle (r = 500 m), are considered as center users. Only the center users can participate consensus process for smooth communication. Although the center vehicles move out from the center circle, they are still inside the RSU coverage.

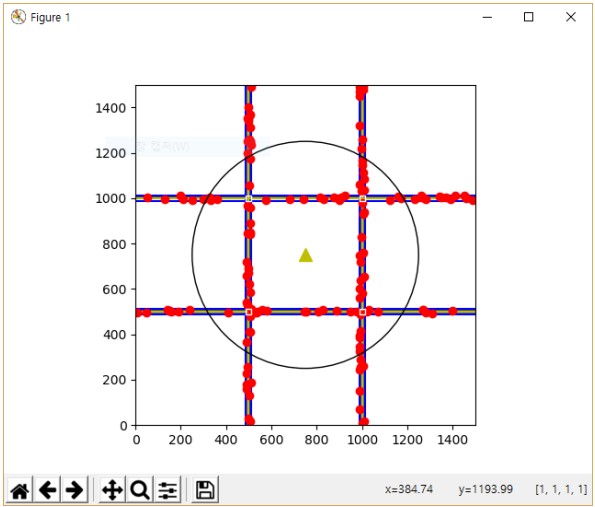

**Figure 12.** Simulation model.

For considering mobility, we refer WINNER 2 B1 pathloss model, shadow fading and several equations for evaluating 802.11p-based vehicular network [23–25]. Figure 13 shows the calculated signal-to-noise ratio (SNR) according to the distance between a transmitter and a receiver at line of sight (LOS) and non-line of sight (NLOS) scenarios, respectively. High SNR means that the quality of the channel is good, so a sender can transmit more data successfully. In the NLOS environment, SNR is lower due to the obstacles. In our model, if two vehicles are located in the same street, LOS path loss model is applied. On the other hand, NLOS path loss model is applied when the vehicles are located in different street.

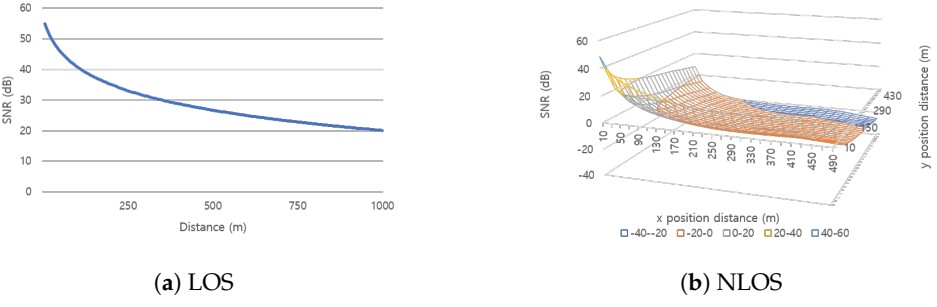

(**a**) LOS          (**b**) NLOS

**Figure 13.** SNR vs Distance.

We also derive SNR values according to the position of vehicles through the simulation as shown in Figure 14. Depending on the positions between a sender and a receiver, SNRs are different. Also, we can see if there are more vehicles, SNR would be higher because the average distance between two vehicles becomes shorter. Finally, we calculate average bit-error rate (BER) upon different modulation schemes such as BPSK 1/2 (3 Mbps), QPSK 1/2 (6 Mbps), 16QAM 1/2 (12 Mbps) and 64QAM 1/2 (24 Mbps) as shown in the Figure 15. In our system, we assume that the average BER is same with packet error rate (PER) as $10^{-4}$ for simplification. Thus, a few messages in consensus may not be received because of the packet error. Each vehicle waits until it receives enough messages to determine whether agree or not. If the consensus is failed, new consensus with a new leader, called as the view change, would occur.

Table 3 shows the total consensus delay of each algorithm. The terms for consensus messaging delay, block transmission delay, singing delay, verification delay, and total consensus delay are defined as $t_m$, $t_{block}$, $t_{sign}$, $t_{ver}$, and $t_{total}$, respectively. The proposed scheme is based on the elliptic curve cryptograph (ECC). The public and private keys are generated from the elliptic curve which is one of asymmetric cryptographic methods. The system key is a symmetric key, so the advanced encryption standard (AES-256) is used for encryption and decryption. Thus, we define signing delay

and verification delay as 7 ms and 23 ms respectively, where the secp384r1 curve is used. In the proposed scheme, the size of a transaction is 240 bytes and messages for consensus is 200 bytes. According to the size, the transmission delay can be derived.

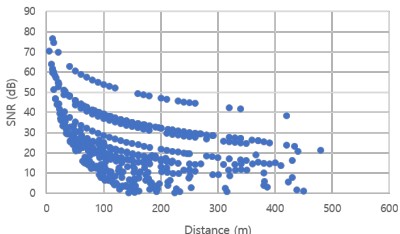

(**a**) 50 vehicles

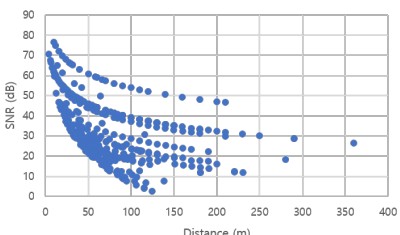

(**b**) 150 vehicles

**Figure 14.** SNR vs Distance by simulation.

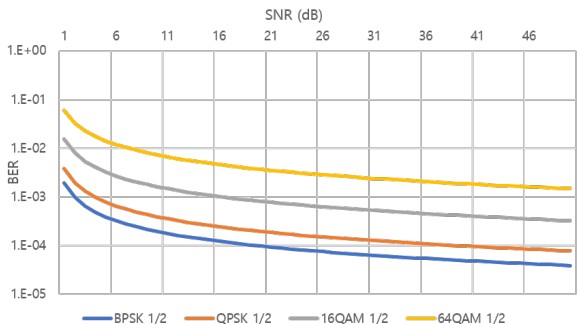

**Figure 15.** Average bit error rate.

**Table 1.** Notations and description.

| Notations | Description |
| --- | --- |
| $Vehicle_i$ | The *i*-th vehicle |
| $BI_i$ | Biological identity of $Vehicle_i$ |
| $k_{LTA_i}$ | A shared system key in the region of $LTA_i$ |
| $Pu_{RTA}$ | Public key of RTA |
| $Pr_{RTA}$ | Private key of RTA |
| $Pu_{Vehicle_i}$ | Public key of *i*-th vehicle |
| $Pr_{vehicle_i}$ | Private key of *i*-th vehicle |
| $prevhash$ | Hash value of previous block |
| $mac_{key}$ | Message authentication code using *key* |
| $ts$ | Time stamp |
| $h(.)$ | Hash function |
| $Enc_{key}(X)$ | Encryped message $X$ with *key* |
| $Dec_{key}(X)$ | Decryped message $X$ with *key* |
| $\|\|$ | Concatenation function |

**Table 2.** Notation and descriptions for Burrows, Abadi, and Needham (BAN) logic.

| Notation | Description |
|---|---|
| $P \mid\equiv X$ | $P$ believes $X$ |
| $P \triangleleft X$ | $P$ sees $X$, or $P$ holds $X$ |
| $P \mid\sim X$ | $P$ had sent $X$ |
| $P \mid\Rightarrow X$ | $P$ completely controls over $X$ |
| $\#(X)$ | X is fresh |
| $\mid\xrightarrow{k} P$ | $P$ uses $k$ as a secret key |
| $P \mid\not\equiv X$ | $P$ cannot believe $X$ |
| $\dfrac{Rule1}{Rule2}$ | If rule1 is true, then rule2 is true |

**Table 3.** Total consensus delay ($n$ = number of vehicles).

| Process | PBFT | LFT | HotStuff |
|---|---|---|---|
| Pre-prepare/Prepare | $t_{sign} + 4n * t_m$ | $t_{sign} + 4n * t_{block}$ | $(n+1)t_{sign} + 4n * t_{block} + 4n * t_m$ |
| Prepare/Pre-commit | $n * t_{ver} + n * t_{sign} + 4n * t_m$ | $n * t_{ver} + n * t_{sign} + 4n * t_m$ | $n * t_{ver} + (n+4) * t_{sign} + 4n * t_m$ |
| Commit | $n * t_{ver} + n * t_{sign} + 4n * t_m$ | $t_{sign} + 4n * t_m$ | $n * t_{ver} + (n+4) * t_{sign} + 4n * t_m$ |
| Reply | $4 * t_{block}$ | - | $n * t_{sign} + 4(n+1) * t_m$ |
| Total | $(2n+1) * t_{sign} + 2n * t_{ver}$ $+12n * t_m + 4 * t_{block}$ | $(n+2) * t_{sign} + n * t + ver$ $+8n * t_m + 4n * t_{block}$ | $(4n+9) * t_{sign} + 2n * t_{ver}$ $+(16n+4) * t_m + 4n * t_{block}$ |

Figure 16 depicts the total consensus delay and transactions per second (tps) for each algorithm when the 20 Kbytes block is applied. We evaluated several block sizes, When the 20 Kbytes block is adopted, 85 transactions can be included in the block. As shown in the Figure 16a, LTF consensus shows relatively lower delay efficient because it reduces one communication process compared with the other algorithms. Hotstuff is an efficient algorithm in wired blockchain network, but pipelining technique cannot be adopted in 802.11p-based vehicular network. Thus, it shows higher delay than the conventional PBFT. On the other hand, Figure 16b shows the transaction/s according to each consensus algorithm. If the participants increase, the more consensus messages should be transmitted for successful consensus. This results in higher delay to confirmation of a block. Consequently, the confirmed transactions (per second) decrease when the more vehicles participate the consensus. In the common scenario in our system, when the 100 vehicles participate consensus, each algorithm can confirm 13, 26 and 11 transactions per second, respectively. In other words, a transaction is confirmed in 76, 38 and 94 ms, and this is tolerable delay to broadcast safety messages in the vehicular network. In addition, the vehicles can check the previous blocks which include confirmed messages, while a new block is verified. Nonetheless, efficiency problem should be continuously considered in the further studies to adopt blockchain to the connected vehicle networks.

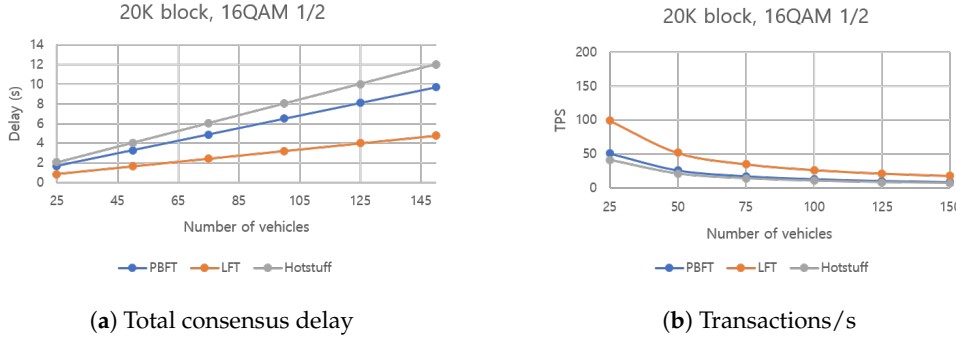

(**a**) Total consensus delay                  (**b**) Transactions/s

**Figure 16.** Consensus delay and TPS.

## 7. Conclusions

In this paper, we propose blockchain-based authentication scheme for connected vehicle network. Most of the conventional schemes are centralized architecture, and didn't consider the attacks from the insider. A few studies adopted blockchain to vehicular network, but most of the studies did not consider mobility issue in vehicular network. In the proposed scheme, we adopt blockchain technology for assuring privacy preservation and efficient authentication. From the blockchain architecture, we can make decentralized network and the vehicles authenticate messages using information in distributed manner without support of the RTA. Through the formal verification and implementation, we show the proposed scheme can achieve several security goals and discuss about the overhead which is caused from the procedures for blockchain. There are several challenging issues in the blockchain-based vehicular network. In the further work, we will consider new consensus algorithm for vehicles which can mitigate the overhead.

**Author Contributions:** Conceptualization, S.J., J.N., S.C.; methodology, S.J.; validation, J.N. and S.J.; formal analysis, J.N.; resources, S.J. and J.N.; writing—original draft preparation, J.N., and S.J.; writing—review and editing, J.N. and S.C.; visualization, J.N. and S.J.; supervision, S.C.; project administration, J.N.; funding acquisition, S.C. All authors have read and agreed to the published version of the manuscript.

**Funding:** This work was supported by the research fund of Signal Intelligence Research Center supervised by the Defense Acquisition Program Administration and Agency for Defense Development of Korea and was supported by the research fund of Hanyang University (HY-2015-N).

**Conflicts of Interest:** The authors declare no conflict of interest.

## Abbreviations

The following abbreviations are used in this manuscript:

| | |
|---|---|
| VANET | Vehicular ad-hoc Network |
| MANET | Mobile ad-hoc Network |
| V2V | Vehicle-to-Vehicle |
| V2I | Vehicle-to-Infrastructure |
| PoW | Proof of Work |
| PoS | Proof of Stake |
| PBFT | Practical Byzantine |
| RTA | Root Trusted Authority |
| LTA | Local Trusted Authority |
| RSU | Road Side Unit |
| OBU | On Board Unit |
| DSRC | Dedicated Short-Range Communication |
| ECDSA | Elliptic curve Digital Signature Algorithm |
| TESLA | Timed Efficient Stream Loss-tolerant Authentication |
| MAC | Message Authentication Code |
| BI | Biological Information |

SNR　　Signal-to-Noise Ratio
LFT　　Loof-fault tolerance
BER　　Bit Error Rate
TPS　　Transaction Per Second

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
