# Peer review of "Distributed Blockchain-Based Message Authentication Scheme for Connected Vehicles"

_electronics, doi:10.3390/electronics9010074_

Round 1

Reviewer 1 Report

The authors proposed the authenitcation process using for composing blockchain system such as the proof of work (PoW) and Practical Byzantine Fault Tolerance (PBFT). The analytical study is really fruitful.  The explanations are good. The proposed method is secure from the internal attacker as well as typical attacks.

The system model is well defined.  The proposed authentication method has various phases such as initiation, registration, message signification, message verification, block generation, and block confirmation.

The analysis and results parts are defined and explained well.

Author Response

 Most of all, we would like to thank the associate editor and the reviewers to spend their valuable and effort to review our paper. This paper proposes a distributed message authentication scheme for autonomous vehicular network using blockchain. After we receive the comments, we modify a few typos and errors and some parts following the other reviewer's comments.

Thank you so much

Reviewer 2 Report

Spelling and grammar mistakes must be corrected, such as “every vehicles can access” in line 39 or “they has system key” in line 306.  In Figure 1, the lines representing V2I and V2V are very similar. Perhaps, the authors could change the style of one of the lines to clarify the figure. Introduction and Related work sections must be improved. Perhaps, in the Introduction, the authors want to clarify a bit more the problem that they are trying to solve and state more clearly their contributions. Readers need to be put in context with some more details of how things currently work and what is briefly the approach of the authors. Authors propose to create a new blockchain scheme for his architecture. In my opinion, the blockchain components should be described more thoroughly. Moreover, it would be better if authors would first justify the need for a new blockchain schema and also justify why existing blockchain solutions, such as Hyperledger, are not suitable in this context. If the new blockchain schema is actually required, then the authors must elaborate more in the solution, compare it with existing solutions and proof that their schema is secure, the vehicles will be able to compute the messages, send and receive even in movement the necessary messages, key distribution will work properly with vehicles in movement constantly connecting and disconnecting to different RSU, etc.  The paper proposes a system using PoW and PBFT, however it is not very clear when and how each consensus mechanism will be used. Moreover, if the paper proposes a new blockchain schema then the related work and the preliminaries sections must be improved regarding this type of systems. Perhaps, authors could include in the preliminaries section some information to help the reader understand how the notation in Section 6 works.   Section 6.3 has some kind of analysis to empirically proof in a simulation that the proposed system works in real life. However, many different measures are mixed in the analysis and it is difficult to visualize why the different plots are included and the relationship with the proposed system. For instance, what is the purpose of Figure 14? SNR and distance were important for the proposed system, that should have been discussed before in some section regarding the design of the system.

Author Response

  Most of all, we would like to thank the associate editor and the reviewers to spend their valuable and effort to review our paper. This paper proposes a distributed message authentication scheme for autonomous vehicular network using blockchain. We try to reflect comments that the editor and the reviewers gave as possible as we can.

Thank you so much.

Reviewer 3 Report

i. the benefits of using blockchain is misplaced in the introduction section. Benefits should be in the results and discussion.

page 53: assure to ensure

page 55: this make ... to this makes

pages 62/63 : In this case, an adversary is hard to penetrate into the communication.  recast --- an adversary finds it difficult to penetrate the communication line ......

page 64: .....used combination of the ECDSA - recast

page 74: ....the [7] categorized--- recast to [7] categorised ....

page 87:    the authors introduce ----- recast to the authors introduced

page 101: a categorical statement such as this, is too early to be made. redact the sentence and possibly put in results and discussion. the statement preempts the probable results ....

page 106: one of the emerging

page: 106-120: Refer to Fig 2, and explain the figure

page 136: Fig 2 mentioned here is far away from where it is situated.

 pages 150-160:  fig 3 should be situated from pages 150-160

2. all figures and diagrams must be referred to and explained on the same page.

page 377: ....cannot judge of sender of a message -  please recast

3. Results are not clearly stated.

4. Results/conclusion/discussion are silent on evaluating this study with previous works done in this area, especially: 

Dorri, A.; Steger, M.; Kanhere, S.S.; Jurdak, R. Blockchain: A distributed solution to automotive security and privacy. IEEE Communications Magazine 2017, 55, 119–125.

Shrestha, R., Bajracharya, R., & Nam, S. Y. (2018, October). Blockchain-based message dissemination in vanet. In 2018 IEEE 3rd International Conference on Computing, Communication and Security (ICCCS) (pp. 161-166). IEEE.

Author Response

(The authors gave the same response as above.)

Round 2

Reviewer 2 Report

 Thank you for you effort.